# Anti-Drug Antibodies in Patients with Inflammatory Bowel Diseases Treated with Biosimilar Infliximab: A Prospective Cohort Study

**DOI:** 10.3390/jcm10122653

**Published:** 2021-06-16

**Authors:** Anna Pękala, Rafał Filip, David Aebisher

**Affiliations:** 1Department of Gastroenterology, IBD Unit of Clinical Hospital 2, Lwowska 60 Str., 35-301 Rzeszow, Poland; apekala@onet.pl; 2Faculty of Medicine, University of Rzeszow, Kopisto 2A Str., 35-315 Rzeszow, Poland; 3Department of Photomedicine and Physical Chemistry, Faculty of Medicine, University of Rzeszow, Warzywna 1A Str., 35-310 Rzeszow, Poland; daebisher@ur.edu.pl

**Keywords:** inflammatory bowel disease, biosimilar infliximab, immunogenicity, ADA, CT-P13, anti-TNF

## Abstract

Reports of the prevalence of antibodies to infliximab (anti-drug antibodies, ADA) are inconsistent due in part to the various assay formats used to monitor immunogenicity in the clinic and under clinical trial settings. This study aimed to determine the frequency of ADA in patients with inflammatory bowel disease (IBD) during induction and maintenance therapy with biosimilar infliximab (CT-P13) using the ELISA (enzyme-linked immunosorbent assay) method. In this prospective single-center study, we analyzed the incidence of ADA and the relationship between the presence of ADA and the following variables: gender, type of disease, immunosuppressive therapy used, and duration of treatment. A total of 84 patients with IBD received CT-P13 and were followed up for an average of 7 months. We found ADA in 50% of the patients with undetectable levels of the drug. The percentage of persons with antibodies detected during induction treatment was 11.3% compared to 9.6% during maintenance therapy. The analysis showed no relationship between response to treatment and antibody titers (*p* = 0.381). The study showed a statistically significant relationship between undetectable levels of CT-P13 and the presence of ADA at week 6 of therapy (i.e., ADA were detected in all the patients with undetectable levels of CT-P13). Patients with IBD and undetectable levels of CT-P13 before administration of the third induction dose were at high risk of the presence of anti-drug antibodies as well as primary non-response.

## 1. Introduction

Infliximab (IFX) is a chimeric monoclonal IgG1 antibody against tumor necrosis factor (TNF) that was introduced for treatment approximately 20 years ago [1]. Currently, its greatest clinical application is in the treatment of inflammatory bowel disease [2]. In 2013, the biosimilar IFX–CT-P13 was approved and is now often used in place of the originator after the patent rights expired [3]. Its introduction has significantly increased the availability of biological therapy for patients with Crohn’s disease and ulcerative colitis [4,5,6]. Although the role of anti-TNF preparations remains significant, in 10–30% of cases there is a primary lack of response, and 23–46% of patients lose response during treatment (secondary loss of response [LOR]) [7]. Measuring the level of infliximab makes it possible to distinguish between patients with normal levels of the drug who require a change of drug and patients with non-therapeutic infliximab levels who require a dose increase. For the second group of patients, the treatment limitation may be immunogenicity. The formation of anti-drug antibodies in patients treated with IFX may arise during both induction and maintenance therapy, resulting in a lack or loss of response [8]. The development of ADA may result in neutralization as well as accelerated drug clearance in non-immune mechanisms, leading to a non-therapeutic level of the drug in the blood serum and consequently reducing the effectiveness of treatment [9]. Anti-drug antibodies also increase the risk of an allergic reaction to the infusion [10]. The significance of this phenomenon has prompted several gastroenterological societies to recommend monitoring of drug and antibody levels [11,12,13]. Monitoring patients during treatment with anti-TNF drugs enables the assessment of the mechanism of loss of response and therefore makes it possible to decide on an appropriate therapeutic intervention [14]. This tool can additionally be used to assess the risk of lack or loss of response, which is especially useful in steroid-dependent patients when the response to biological treatment may be initially ambiguous. Due to the growing number of new types of drugs acting on TNF alpha-independent inflammatory pathways, the selection of patients who can best benefit from a given type of treatment is gaining importance. Monitoring the level of IFX and the presence of anti-drug antibodies, especially at the stage of induction treatment, can be used to predict response and enable faster individualization of therapy including dose modification or switching to another drug [15]. Due to the use of various tests for the determination of various types of antibodies (neutralizing or otherwise), the status of a person with low drug levels and the absence or presence of ADA is not clear-cut and this issue remains under investigation [16,17,18].The purpose of this study was to determine the incidence of ADA in patients with non-therapeutic levels of CT-P13 in clinical practice and to determine risk factors for their occurrence in a group of patients monitored proactively during induction treatment. Additionally, we examined the relationship between CT-P13, ADA levels, and the response to induction treatment, as well as between undetectable levels of CT-P13 and the presence of ADA depending on the duration of treatment.

## 2. Materials and Methods

### 2.1. Study Design

The study included patients with IBD treated with biosimilar infliximab CT-P13 (Remsima) at the tertiary IBD center in south-eastern Poland (city of Rzeszow) between 2016 and 2019.

Before starting treatment, all patients had an endoscopic examination together with an assessment of disease activity using the Mayo score for ulcerative colitis (UC) or the Crohn’s Disease Activity Index (CDAI). Some patients qualified for treatment due to the presence of perianal fistulas.

The study consisted of two groups of patients. Patients in group A were proactively monitored during induction treatment (at week 6 and week 14). Patients in group B were monitored only during maintenance treatment once they presented with clinical signs of secondary LOR (Figure 1). In group A, drug level testing was scheduled before the last induction dose at week 6 (1st measurement) and just before the first maintenance dose in week 14 (2nd measurement). In group B, drug level was tested once the patient manifested LOR. The measurements took place every 4 or 8 weeks starting from week 22. Additionally, ADA were tested in all patients with non-therapeutic drug levels (below 3 μg/mL).

CT-P13 was administered according to the induction schedule at a dose of 5 mg/kg as an intravenous infusion at week 0, week 2, and week 6.

During the maintenance therapy, patients received CT-P13 at a dose of 5–10 mg/kg (CD) or 5 mg/kg (UC) at week 14 and later every 4 or 8 weeks (CD) or every 8 weeks (UC) following the therapeutic programs of the National Health Fund of Poland and the local Summary of Product Characteristics (Figure 2 and Figure 3).

The response to treatment was analyzed at every study visit starting at week 14, before each subsequent infusion. UC was evaluated on a full Mayo scale along with endoscopic examination. A response to treatment was defined as a reduction in disease activity by at least 3 points on the Mayo scale. In Crohn’s disease, the response to treatment was assessed using the CDAI scale. Clinical response was defined as a reduction in CDAI by at least 70 points and by at least 25% from baseline or a reduction in fistula secretion by at least 50%. Secondary non-response was defined as worsening of the Mayo/CDAI score in patients who initially responded to treatment accordingly to the criteria described above.

The primary endpoint was the presence of immunogenicity during induction and maintenance treatment with CT-P13 assessed as detection of ADA. Secondary endpoints were: (1) assessment of response to treatment, primary non-response, or secondary LOR depending on drug level and ADA level and (2) risk assessment of ADA depending on the type of disease, sex, immunosuppressive treatment, and the time of detection of low drug levels.

### 2.2. Immunoassays

All assays were conducted at the same central laboratory (Genloxa, Tczew, Poland). Serum levels of CT-P13 and ADA were evaluated using validated ELISA methods (Matriks Biotek, Turkey) as per the manufacturer’s instructions. The detection level of the CT-P13 test was 0.4 to 20 μg/mL, while the adopted therapeutic range was 3 to 7 μg/mL [19]. The anti-drug antibody test was considered positive when the level detected was >2 AU/mL, as per the manufacturer’s instructions.

### 2.3. Statistical Analysis

The mean drug levels in 3 patient groups, depending on treatment response (responders, primary non-responders, and secondary LOR), were compared using the Kruskal–Wallis test. In order to determine how groups differed from each other, Dunn’s post hoc test was conducted. Categorical variables were compared using Fisher’s exact test.

### 2.4. Ethics

The study was approved by the Ethics Committee of the University of Rzeszow (No. 9/10/2016). Each participant read and signed a written informed consent form.

## 3. Results

### 3.1. Patients

The study group included 84 biologic-naïve patients (46 with CD and 38 with UC) treated with biosimilar infliximab CT-P13 (Remsima). The level of disease activity in UC was in the range of 7–12 points on the Mayo score (moderate to severe disease), and Crohn’s disease (CD) activity was in the range of 150–450 points. The duration of treatment ranged from 6 weeks to 12 months (mean 28 weeks). Most of the patients (77 of 84) received combined immunosuppressive treatment with thiopurines (91.7%). Fifty-six patients received additional mesalazine, and 40 patients were also initially treated with glucocorticosteroids. Patient baseline characteristics are shown in Table 1.

### 3.2. Drug Levels and Response to Treatment

In 40 of 84 patients (47.6%), serum trough levels of CT-P13 were higher than or equal to 3 µg/mL throughout the study period. Non-therapeutic drug levels (below 3 µg/mL) were found in 44 of 84 patients (52.4%), and among them, 16 had undetectable drug levels (<0.4 µg/mL). Drug levels below 3 µg/mL were found in 8 of 53 group A patients at week 6, including 6 that were undetectable, and in 36 of 84 patients in groups A and B after week 6, including 10 that were undetectable.

Forty-five of 53 group A patients (84.9%) responded to induction therapy, and 8 of 53 patients (15.1%) showed no response to treatment. Out of these eight patients, who were non-responsive, six had non-therapeutic drug levels (<0.4–2.5 µg/mL), and two patients had drug levels above 7 µg/mL. Among patients who responded to treatment, there was one who had undetectable drug levels at week 6. In group B, all the patients had CT-P13 levels below 3 µg/mL.

Mean drug levels were compared in the three patient groups according to their response to treatment: patients with response to induction treatment (*n* = 45), patients with no such response (*n* = 8), and patients with secondary non-response (*n* = 31; group B). In the group of patients responding to induction treatment, the median drug level at week 6 was 16.70 µg/mL, while in patients with primary non-response it was −0.95 µg/mL. In patients with a secondary non-response, the median drug level was 0.80 µg/mL. Patients with response to induction treatment had a significantly higher average drug level than patients with primary lack of response (*p* = 0.001) or with secondary loss of response (*p* < 0.001). There was no significant difference in the mean drug levels between patients with primary lack of response and those with secondary loss of response (*p* = 0.769).

We also analyzed the drug levels in group A patients with severe disease activity. In group A, seven patients had severe UC and none had severe CD (CDAI above 450 points). Among the seven patients with severe UC, one had undetectable drug levels, was positive for anti-drug antibodies, and manifested lack of response to induction treatment. It is possible that increasing the drug dose would yield clinical benefits; however, in this study the dose was not increased during the induction phase. The remaining six patients with severe UC had therapeutic drug levels and responded to treatment.

### 3.3. Immunogenicity of CT-P13

Anti-drug antibodies were detected in 9 of 84 patients (10.7%): in 6 during the induction treatment (11.3%) and in 3 during the maintenance therapy (9.6%). There was no statistically significant difference in antibody detection during induction treatment as compared to maintenance therapy (*p* = 0.158 in both groups). All of the patients with ADA had non-therapeutic drug levels, including eight with drug levels below the detection limit. Accordingly, the incidence of ADA in patients with non-therapeutic drug levels was 9/44 (20.4%), while in patients with undetectable drug levels it was 8/16 (50%). The results are shown in Table 2.

Among six patients with ADA detected during the induction phase, no response to induction treatment was seen in three patients with a high antibody titer (range from 27.9–30 AU/mL) and two patients with a low antibody titer (range from 2.3–4.2 AU/mL). The remaining one patient responded to treatment despite the presence of ADA. Among three patients with ADA detected during the maintenance phase, LOR occurred in two patients with high antibody titers (>50 U/mL) and one patient with low antibody titer (3.2 AU/mL). The relationship between lack of response and high/low antibody titers is shown in Table 3. The analysis showed no relationship between response to treatment and antibody titers (p = 0.381).

Antibodies were detected in four patients with UC and five patients with CD. All patients with detected ATIs simultaneously received immunosuppressive therapy with thiopurines (two of them at a low dose due to intolerance).

To assess risk factors, the relationships between the presence of antibodies, selected variables (sex, type of disease, duration of treatment, immunosuppressive therapy), and undetectable levels of CT-P13 at 6 weeks were checked. There was no relationship found between sex, type of disease, and lack of immunosuppressive treatment and the presence of ATIs. There was also no difference found between the induction and maintenance treatment group in terms of the occurrence of ADA. At week 6, a statistically significant relationship between undetectable levels of CT-P13 and the presence of antibodies was found (*p* < 0.001), i.e., undetectable levels of CT-P13 occurred in all the patients with antibodies (*n* = 6) and in none of the patients without antibodies. The results are shown in Table 4.

## 4. Discussion

The problem of immunization caused by anti-TNF drugs that makes therapy ineffective has been the subject of many clinical trials, which predominantly studied the maintenance phase. Moreover, the analyses differ in terms of the method of ADA detection making comparisons inconclusive [20,21,22]. In this study, we evaluated the presence and risk factors of ADA during induction and maintenance therapy and used the validated, easily accessible ELISA test to determine drug and ADA levels. Anti-drug antibodies were present in 20.4% of patients with non-therapeutic CT-P13 levels and 50% of patients with undetectable CT-P13 concentrations. Only one patient with detectable drug levels had antibodies simultaneously. The use of a more sensitive test would probably allow for a more accurate assessment. Samples containing the drug should be considered ambiguous and not negative for ADA, and, thus, the percentage of patients with a positive antibody result could be underestimated.

A much larger study was conducted in Canada by Vande Casteele et al. who used a thorough liquid phase shift test to examine a group of 483 patients during maintenance treatment with infliximab. They showed that for 22.9% of patients with undetectable levels of infliximab, as much as 71.8% of patients were positive for ADA [23]. Similarly, our study showed the presence of ADA in 50% of patients with undetectable levels of CT-P13, but with the difference that the tested patients included those in the induction phase of treatment. Considering only samples from patients during maintenance treatment, we found antibodies in 2 out of 10 patients who had undetectable CT-P13 (20%). Conversely, in the group of patients examined during induction treatment (at 6 weeks), antibodies were found in six out of six patients with undetectable CT-P13 (100%). Five out of six patients simultaneously had no clinical and endoscopic response. Considering that there is no masking of the result by the drug present in the serum, the detection of antibodies during maintenance treatment in our study population is relatively low at 20% compared to 71.8% as reported by Vande Casteele et al. The difference may be due to the much smaller sample size of our group and a different type of test used. In other studies where the same test was used, the detection of antibodies was similar, e.g., 22.2% in a group of Brazilian patients [24]. Similar results were also obtained in a Canadian study using a different type of test [25].

An interesting observation made in our study is that the undetectable level of CT-P13 at week 6 was in each case associated with the detection of ADA. If at week 6 drug levels were low (but detectable) and undetectable levels of CT-P13 were found for the first time after induction treatment, the risk of detecting antibodies was lower.

Antibody levels did not determine the response because primary non-response or secondary LOR occurred at both high and low antibody titers. Several studies also showed that anti-infliximab antibodies found at week 6 were associated with primary nonresponse and mucosal healing impairment [26,27]. In our study, the lack of response was evident in most patients by week 6. To our knowledge, none of the studies described such a close relationship between antibody detection at week 6 and lack of response. This relationship was not as pronounced, probably due to the use of more sensitive tests [27,28]. Another important observation is that all patients with antibodies (regardless of antibody titers) who received a third induction dose of CT-P13 had no allergic reaction to the infusion.

As for risk factors for the presence of antibodies, it was not possible to demonstrate the effect of sex, type of disease, or immunosuppressive treatment on ADA production, although such a relationship was observed in other studies [28,29,30,31,32]. In contrast, we noted the relationship between CT-P13 concentration and clinical results patients with response to induction treatment had a much higher average level of the drug compared to patients with primary or secondary non-response. Other studies present similar conclusions [33]. There was no significant difference in mean drug levels between patients with primary and secondary non-response.

The main limitations of this study are the relatively small size of the group and the lack of assessment of the stability of the antibodies detected. Although the use of the ELISA method in detecting ADA is not perfect, other studies have shown that more sensitive tests can also detect antibodies that have no neutralizing potential or are transient and therefore not of clinical significance [34,35]. The advantage of our study is its prospective nature and coverage of patients undergoing induction therapy as fewer studies have been carried out during this period. Furthermore, our study presents data for CT-P13, whereas most other studies present data for original IFX [16,20,21,22,23,24,25,26,27,28,29,30,31,32,33,34,35].

This single-center study, which reflects actual clinical practice, confirms that measuring levels of infliximab during the induction phase is useful because as many as 11.3% of patients had undetectable drug levels and the presence of ADA. In summary, therapeutic drug monitoring (TDM) might help to optimize pharmacological therapy in IBD patients. Undetectable levels of CT-P13 infliximab before administration of the third induction dose may reflect the presence of ADA and the risk of primary non-response. However, ADA titers were not critical to treatment response and no statistically significant relationship was found between primary and secondary loss of response and ADA titers.

## 5. Conclusions

Therapeutic drug monitoring (TDM) might help to optimize pharmacological therapy in IBD patients. Undetectable levels of CT-P13 infliximab before administration of the third induction dose may reflect the presence of ADA and the risk of primary non-response. However, ADA titers were not critical to treatment response and no statistically significant relationship was found between primary and secondary loss of response and ADA titers.

## Figures and Tables

**Figure 1 jcm-10-02653-f001:**
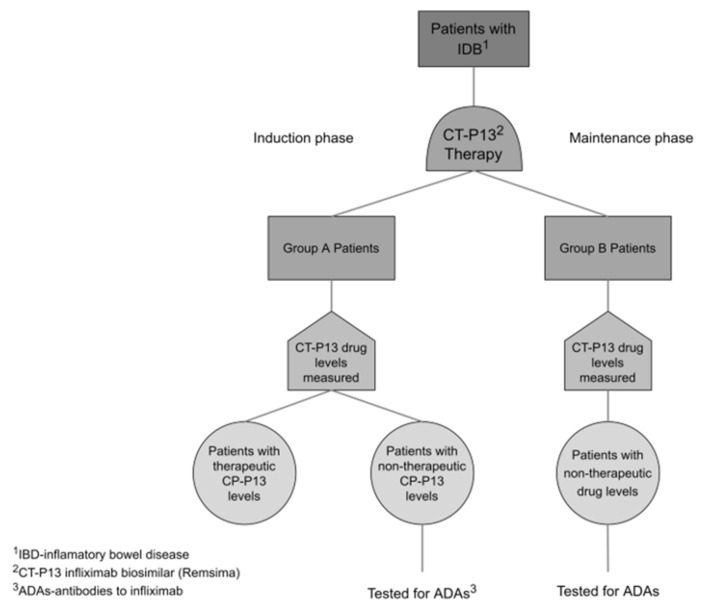
Study Design.

**Figure 2 jcm-10-02653-f002:**
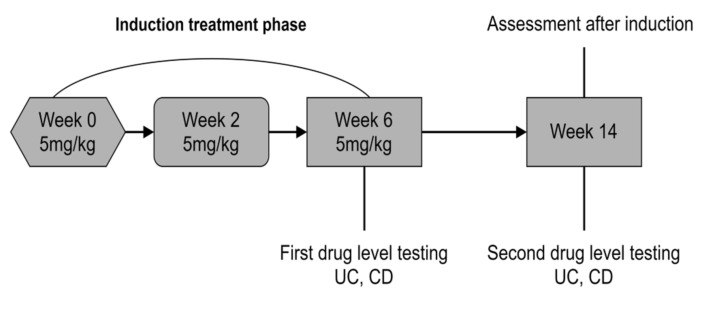
CT-P13 dosing and drug level testing schedule in group A.

**Figure 3 jcm-10-02653-f003:**
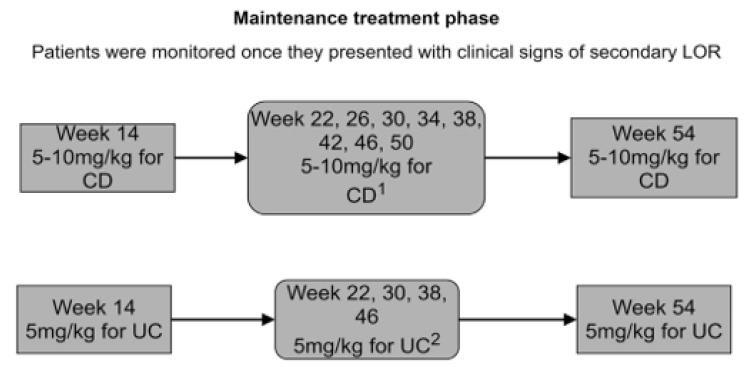
CT-P13 dosing and drug level testing schedule in group B.

**Table 1 jcm-10-02653-t001:** Patient characteristics.

	**Group A**	**Group B**
***n***	**53**	**31**
Females, *n* (%)	27 (50.9)	18 (58.0)
Males, *n* (%)	26 (49.0)	13 (41.9)
Age, median years (range)	35.5 (18–68)	29.5 (19–52)
Age at diagnosis, median years (IQR)	5.5 (1–12)	4.5 (1–9)
Smoking status, *n* (%)		
Non-smoker	50 (94.3)	27 (87.1)
Former smoker	2 (3.8)	3 (9.7)
Current smoker	1 (1.9)	1 (3.2)
***Concomitant Treatment***		
Thiopurines, *n* (%)	49 (92.4)	28 (90.3)
Steroids, *n* (%)	38 (71.7)	2 (6.5)
Mesalazine, *n* (%)	38 (71.7)	18 (58.0)

***Crohn’s Disease***		**Group A/Group B (*n*)**
Age at diagnosis, *n*	A1 (diagnosed <17 years of age)	2/3
	A2 (diagnosed 17–40 years of age)	15/15
	A3 (diagnosed >40 years of age)	8/3
Disease location, *n*	L1 (ileal)	4/4
	L2 (colonic)	10/5
	L3 (ileocolonic)	13/7
	L3 + (ileocolonic) + L4 (upper gastrointestinal tract)	1/2
Disease behavior, *n*	B1 (non-stricturing, non-penetrating)	20/15
	B2 (stricturing)	0/1
	B3 (penetrating)	5/5
CDAI at the start (average)	150–450p (357.8)	150–381p (305.2)/180–450p (340.5)
***Ulcerative Colitis***		**Group A/Group B (*n*)**
Extent, *n*	E1 (proctitis)	2/1
	E2 (left-sided colitis)	18/5
	E3 (pancolitis)	8/4
Severity, *n*	S1 (mild)	3/3
	S2 (moderate)	18/6
	S3 (severe)	7/1
Mayo score at the start (average)	7–12p (8)	7–12p (8)/7–12p (8)

**Table 2 jcm-10-02653-t002:** Relationship between CT-P13 trough concentration and ADA status.

CT-P13	Number of Patients *n* = 84	Number of Patients with Antibodies *n* = 9
CT-P13 levels tested during induction treatment at week 6 and at week 14	53 (group A)	6/53 (11.3%)
CT-P13 levels tested during maintenance therapy (after week 14)	31 (group B)	3/31 (9.6%)
Level < 3 μg/mL (Including undetectable)	44/84 (53%) 16/44 (36.4%)	9/44 (20.4%) 8/16 (50%)
Determination time for CT-P13 < 3 μg/ml		
week 6	8/53 (15.1%)	6/8 (75%)
>week 6	36/84 (42.8%)	3/36 (8.3%)
Undetectable CT-P13 at 6 weeks	6/53 (11.3%)	6/6 (100%)
Undetectable CT-P13 > 6 weeks	10/84 (11.9%)	2/10 (20%)

**Table 3 jcm-10-02653-t003:** Relationship between response to treatment and antibody titers.

	High Antibody Titer (*n* = 5)	Low Antibody Titer (*n* = 4)	*p*-Value
**Response to Induction Treatment**	0 (0.0%)	1 (25.0%)	0.381
**Primary Non-Response**	3 (60.0%)	2 (50.0%)	
**Loss of Response**	2 (40.0%)	1 (25.0%)	

**Table 4 jcm-10-02653-t004:** Analysis of potential risk factors for the presence of ADA.

**Factor**	**The Whole Group**	**Antibodies**	**No Antibodies**	***p*-Value**
***n***	**84**	**9**	**75**	
**Sex**				
Female	45 (53.6%)	4 (44.4%)	41 (54.7%)	0.727
Male	39 (46.4%)	5 (55.6%)	34 (45.3%)	
**Disease**				
Ulcerative colitis	38 (45.2%)	4 (44.4%)	34 (45.3%)	>0.999
Crohn’s disease	46 (54.8%)	5 (55.6%)	41 (54.7%)	
**Treatment Time**				
During induction	53 (63.1%)	6 (66.7%)	47 (62.7%)	>0.999
During maintenance treatment	31 (36.9%)	3 (33.3%)	28 (37.3%)	
**Immunosuppressive Treatment**				
Yes	77 (91.7%)	9 (100.0%)	73 (97.3%)	>0.999
No	7 (8.3%)	0 (0.0%)	2 (2.7%)	
**Undetectable Infliximab Level at 6 weeks ***				
Yes	6 (11.3%)	6 (100%)	-	<0.001
No	47 (88.7%)	-	47 (62.7%)	

* Data given as *n* (% of the subgroup). Fisher’s exact test *N* = 53.

## Data Availability

Data presented in this study are available on request from Rafał Filip and Anna Pękala. The data are not publicly available due to privacy restrictions.

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
