# Peer review of "Anti-Drug Antibodies in Patients with Inflammatory Bowel Diseases Treated with Biosimilar Infliximab: A Prospective Cohort Study"

_jcm, 2021, doi:10.3390/jcm10122653_

Round 1
Reviewer 1 Report
One of the limitations of anti-TNF biologicals (both originator and biosimilar drugs) is loss of response in up to 30–50% of patients with or without the development of anti-drug antibodies (DA). Reports published to data show that the prevalence of ADA to anti-TNF biologicals vary between studies, in part due to inconsistencies in design and performance of drug-tolerant ADA assays (e.g. exclusive detection of free and not target-bound ADA), which places more importance on measurement of the serum drug trough levels (sometimes in combination with the ADA). Such approach was demonstrated for IFX by Martis et al. (J Crohns Colitis. 2014 Aug; 8(8):881-9), but generally studies attempting to correlate these outcomes with clinical improvement with treatment, especially with biosimilars, are lacking and remain in demand.
In this manuscript, Pekala et al. monitored the relationship between ADA and the level of serum CT-P13 IFX biosimilar in UC and CD patients during the induction and maintenance phases of the treatment. They monitored standardized response to treatment during each phase and associated it with the serum drug level and the presence a titer of the ADA’s. The study was performed with acceptable level of rigor, is reasonably powered, and the results are interpreted with caution and with careful consideration of previous work.
Minor comments:
The abbreviation ATI is typically used in reference to anti-infliximab antibodies. Since the focus of the manuscript is a biosimilar, I would suggest the use of ADA acronym for anti-drug antibodies, also used commonly in the literature.
The title of the manuscript clearly suggests a significant correlation of anti-drug antibodies (ADA) with the CT-P13 trough level, but the data to back it up is not that clear. There are considerable differences in the proportions of patients with sub-therapeutic concentration of CT-P13 and those with ADA (per Table 2). It is plausible that low CT-P13 levels are better predictors of non-response than the presence of ADA, and that could be part of the analysis in Table 3.
Reviewer 2 Report
-I would add a table showing baseline characteristics of patients in Group A and B as the results are focused in these 2 groups and you are not comparing MC vs UC patients.
-It would be interesting if you had 6TGN levels too to see if in patients with development of ATIs is related with 6TGN levels.
-I would maybe add a comment about which patients in group A who had low levels of CT-P13 had a severe disease activity (probably needed elevated induction dose of anti-TNF).
